# High BMI1 Expression with Low CD8+ and CD4+ T Cell Activity Could Promote Breast Cancer Cell Survival: A Machine Learning Approach

**DOI:** 10.3390/jpm11080739

**Published:** 2021-07-28

**Authors:** Yumin Chung, Kyueng-Whan Min, Dong-Hoon Kim, Byoung Kwan Son, Sung-Im Do, Seoung Wan Chae, Mi Jung Kwon

**Affiliations:** 1Department of Pathology, Kangbuk Samsung Hospital, Sungkyunkwan University College of Medicine, Seoul 03181, Korea; yumin.chung@samsung.com (Y.C.); sungim.do@samsung.com (S.-I.D.); swan.chae@samsung.com (S.W.C.); 2Department of Pathology, Hanyang University Guri Hospital, Hanyang University College of Medicine, Guri 11923, Korea; 3Department of Internal Medicine, Uijeongbu Eulji Medical Center, Eulji University School of Medicine, Uijeongbu 11749, Korea; sbk1026@eulji.ac.kr; 4Department of Pathology, Hallym University Sacred Heart Hospital, Hallym University College of Medicine, Anyang 14068, Korea; mulank@hanmail.net

**Keywords:** BMI1, breast neoplasms, survival, gene, machine learning

## Abstract

BMI1 is known to play a key role in the regulation of stem cell self-renewal in both endogenous and cancer stem cells. High BMI1 expression has been associated with poor prognosis in a variety of human tumors. The aim of this study was to reveal the correlations of BMI1 with survival rates, genetic alterations, and immune activities, and to validate the results using machine learning. We investigated the survival rates according to BMI1 expression in 389 and 789 breast cancer patients from Kangbuk Samsung Medical Center (KBSMC) and The Cancer Genome Atlas, respectively. We performed gene set enrichment analysis (GSEA) with pathway-based network analysis, investigated the immune response, and performed in vitro drug screening assays. The survival prediction model was evaluated through a gradient boosting machine (GBM) approach incorporating BMI1. High BMI1 expression was correlated with poor survival in patients with breast cancer. In GSEA and in in silico flow cytometry, high BMI1 expression was associated with factors indicating a weak immune response, such as decreased CD8+ T cell and CD4+ T cell counts. In pathway-based network analysis, BMI1 was directly linked to transcriptional regulation and indirectly linked to inflammatory response pathways, etc. The GBM model incorporating BMI1 showed improved prognostic performance compared with the model without BMI1. We identified telomerase inhibitor IX, a drug with potent activity against breast cancer cell lines with high BMI1 expression. We suggest that high BMI1 expression could be a therapeutic target in breast cancer. These results could contribute to the design of future experimental research and drug development programs for breast cancer.

## 1. Introduction

Breast cancer is the most frequently diagnosed cancer and the leading cause of cancer-related death among women worldwide, as reported by the GLOBOCAN 2018 database published by the International Agency for Research on Cancer. GLOBOCAN 2018 provides information on the estimated incidence and mortality rates in 185 countries for a total of 36 types of cancer [1]. In the breast, there are different histological types of malignancy such as ductal carcinoma, lobular carcinoma, medullary carcinoma and micropapillary carcinoma. As a mesenchymal-type, breast cancer originating from stromal cells may occur rarely [2].

Molecular/genetic studies and evaluation of targeted therapies for breast cancer have been actively conducted to improve patient survival, but these approaches are still seen as a challenge by many clinicians. Therefore, it is necessary to continuously discover carcinogenic factors with the aim of developing new therapeutic approaches.

Immunotherapy has been recognized as an innovative treatment approach for many types of solid tumors, indicating how important it is to understand the exact contribution of the immune system to the response to cancer treatment. Immune checkpoint inhibitors targeting cytotoxic T-lymphocyte antigen-4 (CTLA-4), programmed cell death protein 1 (PD-1), and programmed cell death ligand-1 (PD-L1) have shown positive results in clinical trials [3,4]. Anti-PD-1 antitumor immunity can be enhanced by elevated numbers of CD8+ T cells that secrete cytotoxic substances and interferon gamma (IFN-γ) [5]. Several immunotherapeutic drugs are Food and Drug Administration (FDA) approved for other solid tumors, but no such drugs have been approved for breast cancer [6]. A previous study reported that a robust immune response could improve the survival of patients with breast cancer [7,8]. One study demonstrated that increased tumor-infiltrating lymphocytes (TILs) were associated with favorable prognosis in patients with breast cancer [7]. In addition, another study reported that PD-L1 was frequently expressed in human epidermal growth factor receptor 2 (HER-2)-positive or triple-negative subtypes compared to luminal subtypes [8]. There are several biomarkers that has been discussed as prognostic factors such as JAK1, ITGA3, and MacroH2A1 in relation to the immune activity of breast cancer [9,10,11].

B-lymphoma Moloney murine leukemia virus insertion region-1 (BMI1), a polycomb group family member, is an essential transcriptional repressor and plays an important role in the regulation of stem cell self-renewal in both endogenous and cancer stem cells [12]. Many studies have reported that high BMI1 expression is associated with poor prognosis in different types of malignancies, such as pancreatic adenocarcinoma, hepatocellular carcinoma, prostate cancer, nasopharyngeal cancer, endometrial adenocarcinoma, acute myeloid leukemia, and chronic myeloid leukemia [13,14,15,16,17,18,19,20].

In a meta-analysis, high BMI1 expression was found to be related to poor overall survival (OS) in the Asian group but to favorable OS in the Caucasian group [21]. A previous study revealed high BMI1 expression in high-grade breast cancer tissues compared to normal breast tissues. Furthermore, low BMI1 expression was found to enhance the chemosensitivity of cancer cells [22]. Another study demonstrated that upregulation of BMI1 was associated with gene sets linked to decreased immune activity [23]. Another study reported that BMI1 inhibition could enhance endogenous immune responses by CD8+ T cells [24].

The aim of this study was to evaluate BMI1 expression and to analyze clinicopathological parameters and survival according to BMI1 expression in our cohort of patients with breast cancer. Validation of the survival analysis results was performed with data from The Cancer Genome Atlas (TCGA) database. In various machine learning algorithms [25,26,27], we analyzed the effect of BMI1 on the survival of breast cancer patients using the gradient boosting machine (GBM) algorithm [28]. We evaluated gene sets related to BMI1 using gene set enrichment analysis (GSEA) and pathway-based network analysis of data in the TCGA database [29,30,31]. Furthermore, antitumor immune responses according to BMI1 expression were investigated depending on the counts of CD8+ and CD4+ T cells. Using the Genomics of Drug Sensitivity in Cancer (GDSC) and the Catalog of Somatic Mutations in Cancer (COSMIC) databases, we performed high-throughput drug sensitivity screening in breast cancer cell lines according to BMI1 expression [32,33] (Figure 1).

## 2. Materials and Methods

### 2.1. Patient Selection

A total of 389 patients with invasive ductal carcinoma (IDC) who had undergone surgery at Kangbuk Samsung Medical Center (KBSMC) in Korea between 2005 and 2015 were enrolled in this study. Our study was designed with reference to the Reporting Recommendations for Tumor Marker Prognostic Studies criteria [34]. We evaluated the following parameters: survival rate, age, tumor stage, histological grade, lymphovascular invasion, perineural invasion, estrogen receptor (ER) status, progesterone receptor (PR) status, and HER2 status. We obtained survival rates using data from TCGA, including data for 789 patients who were diagnosed with IDC.

This study protocol was approved by the Institutional Review Board of Kangbuk Samsung Medical Center (IRB number: 2021-04-021).

### 2.2. Tissue Microarray Construction and Immunohistochemistry

Tumor tissue microarrays were constructed from specimens using a tissue arrayer (AccuMax Array; ISU ABXIS Co. Ltd., Seoul, Korea). The tumor tissue microarrays consisted of 3.0-mm tissue cores from representative paraffin blocks. Redundant tissue cores were used in each donor block, taking into consideration that there are limitations in establishing a representative area of the tumor. The percentage of tumor tissue in the tissue cores was >70%. Immunostaining was performed on 4-μm tissue sections using a Bond Polymer Refine Detection System (Leica Biosystems Newcastle Ltd., Newcastle, UK) in accordance with the manufacturer’s instructions. The antibodies used were as follows: anti-BMI1 (1:100 ab14389, Abcam, Cambridge, UK), anti-ER (1:200, Lab Vision Corporation, Fremont, CA, USA), anti-PR (1:200, Dako, Glostrup, Denmark), and anti-HER2 (1:1, Ventana Medical Systems inc., Tucson, AZ, USA).

Evaluation of immunohistochemical staining was carried out by estimating the staining intensity and the proportion of positive cells (Figure 2A). Semiquantitative analysis of the stained sections was conducted by light microscopy using the immunoreactive score (IRS) defined by Remmele et al., which evaluates the proportion of positive cells and the strength of staining [35]. Receiver operating characteristic (ROC) curve analysis was performed to determine the cutoff point for BMI1 expression. BMI1 expression was classified as low (IRS < 1) or high (IRS ≥ 1).

### 2.3. Gene Sets, In Silico Flow Cytometry, and Network-Based Analysis

Data for a total of 789 patients with IDC with available RNA-Seq data were obtained from TCGA [36]. We examined gene sets using GSEA software (version 4.1.0) developed at the Broad Institute at MIT. A gene set comprising 4872 genes with immunologic signatures was applied to find gene sets linked to high BMI1 expression [37]. For this analysis, 1000 permutations were utilized to calculate p values, and the permutation type was set to “phenotype”. Significant gene sets were identified as those with a *p*-value ≤ 0.05, false discovery rate (FDR) < 0.2, and family-wise error rate (FWER) ≤0.4. We applied in silico flow cytometry to determine the proportions of 22 subgroups of immune cells, taking 547 genes into account [31]. We performed molecular pathway-based network analysis using Cytoscape (version 3.8.2) to visualize pathway annotation networks [30,38].

### 2.4. Machine Learning Algorithm

We integrated BMI1 expression with clinical risk factors (T stage, N stage, histological grade, lymphatic invasion, ER status, and HER2 status) to construct composite prognostic models for survival prediction by applying machine learning (ML) algorithms in 389 cases (randomization: training set, 70%; validation set, 30%). A learning algorithm was independently applied to select and combine multiple covariates in the GBM model based on multivariate Gaussian models. In this step, the ”forward” search method, which initiates on a prototype set and selects a feature if and only if the addition of the feature could increase the performance of the prognostic model, is adopted for sequential selection of optimal features. Hyperparameters of ML algorithms, such as the learning rate in the GBM approach, were optimized for each combination of selected covariates and learning algorithm by grid search cross-validation over a predefined range. We performed a search across 54 models with varying learning rates and tree depths. The final optimal models were trained based on the selected covariates and the optimized hyperparameters [28]. To explore the performance of the GBM model, ROC analysis was used.

### 2.5. Data Extraction from the GDSC and COSMIC Databases

Drug screening was performed using datasets from the GDSC and COSMIC databases, large-scale cancer cell line and drug response databases containing data for 809 cancer cell lines and 198 compounds, respectively. Anticancer drug sensitivity was measured in 13 breast cancer cell lines by determining the natural log of the half-maximal inhibitory concentration (LN IC50). A drug was defined as an effective BMI1-targeted drug when the LN IC50 was decreased in breast cancer cell lines with high BMI1 expression but increased in those with low BMI1 expression.

### 2.6. Statistical Analysis

The χ^2^ test and a linear-by-linear association test were used to compare clinicopathological parameters between the high and low BMI1 expression groups. Student’s *t*-test and/or Pearson correlation analysis were used to examine differences between continuous variables. The Breslow test and multistep Cox regression analysis were performed to analyze the relationships between the BMI1 expression level and the following survival rates: disease-free survival (DFS), defined as survival from the date of diagnosis to recurrence/new distant metastasis; and disease-specific survival (DSS), defined as survival from the date of diagnosis to cancer-related death. Statistical significance was considered when *p*-values were ≤ 0.05. Statistical analyses were conducted using the R packages and SPSS statistical software program (version 25.0; SPSS Inc., Chicago, IL, USA).

## 3. Results

### 3.1. Clinicopathologic Characteristics and Survival Rate of Patients with Breast Cancer Stratified by BMI1 Expression

We analyzed the relationships between BMI1 expression and survival rates for 389 and 789 patients with IDC from the KBSMC cohort and TCGA, respectively. In the KBSMC cohort, 225 patients showed low BMI1 expression (57.8%), and 164 had high BMI1 expression (42.2%). High BMI1 expression was frequently observed in the ER- and PR-negative groups (*p* < 0.001 and =0.015, respectively) (Table 1).

In survival analyses of the KBSMC cohort, patients with high BMI1 expression showed significantly shorter DFS and DSS times than those with low BMI1 expression (*p* = 0.029 for DFS and 0.015 for DSS) (Figure 2B). In survival analyses of patients with the luminal, HER2-positive, and triple-negative subtypes, high BMI expression was associated with worse DFS and DSS only in patients with the luminal subtype (*p* = 0.040 for DFS and 0.012 for DSS). There was no relationship between BMI1 expression and DFS/DSS in patients with the HER2-positive and triple-negative subtypes. After adjustment for confounders, including T stage, N stage, histological grade, lymphatic invasion, ER status, PR status, and HER2 status, high BMI1 expression was still related to worse DFS and DSS (Table 2) (*p* = 0.028 for DFS and  0.019 for DSS). In patients with the luminal subtype, high BMI expression was associated with poor DFS and DSS (*p* = 0.050 for DFS and  0.017 for DSS).

In the TCGA database, BMI1 expression was elevated in primary tumor tissues compared to normal tissues (*p* < 0.001) (Figure 2C). We analyzed the relationships between BMI1 and both DFS and DSS. High BMI1 expression was correlated with poor DSS (*p* = 0.033). High BMI1 expression showed a tendency to be related to shorter DFS (*p* = 0.072) (Figure 2D).

### 3.2. Gene Set Enrichment Analysis, Anticancer Immune Response, and Network-Based Analysis of BMI1

We identified gene sets related to decreased CD8+ T cell and CD4+ T cell fractions in the high BMI1 expression group in the TCGA database: GSE44649, naïve versus activated CD8+ T cell, mir155 knockout downregulation and GSE45739, N-ras knockout versus wild-type unstimulated CD4+ T cell downregulation (Figure 3A).

On the basis of the GSEA results, we analyzed the association between BMI1 expression and markers related to each gene set. High BMI1 expression was related to a decreased CD8+ T cell fraction and activated memory CD4+ T cell fraction, a decreased TIL percentage, and reduced expression of CD274 (encoding PD-L1, programmed death-ligand 1) (*p* = 0.018, <0.001, <0.001, and =0.011, respectively) (Figure 3B).

### 3.3. Pathway-Based Network Analysis, Machine Learning and Drug Screening

We performed a molecular interaction pathway-based network analysis on the basis of the expression of BMI1 and immune-related genes (Figure 4). BMI1 was directly linked to the following representative enriched gene ontology (GO) terms and pathways: (1) transcriptional regulation by E2F6, and (2) polycomb repressive complex 1 (PRC1, hPRC-H). Furthermore, BMI1 was indirectly linked to 5 representative functionally enriched GO terms and pathways: (1) inflammatory response pathway, (2) PD-L1 expression and PD-1 checkpoint pathway in cancer, (3) antigen processing and presentation, (4) T cell receptor signaling pathway, and (5) heterotypic cell-cell adhesion.

We compared the performance of the two GBM models for predicting survival rates (Model 1: T stage, N stage, histological grade, lymphatic invasion, ER status, and HER2 status versus Model 2: BMI1 expression level, T stage, N stage, histological grade, lymphatic invasion, ER status, and HER2 status). ROC curves were generated (areas under the curve: Model 1, 0.727; Model 2, 0.736) (Figure 5A,B). We found that the GBM algorithm had the best performance, while the addition of BMI1 to the prediction model improved its prognostic performance.

We identified the 4 anticancer drugs that most effectively reduced the growth of cell lines with high BMI1 expression: telomerase inhibitor IX, nilotinib, tamoxifen, and RO-3306. Cell lines with high BMI1 expression were highly sensitive to telomerase inhibitor IX (Figure 5C).

## 4. Discussion

BMI1 could play a critical role in the development of breast cancer. A study by Paranjape et al. reported that BMI1 overexpression was observed in 64% of IDC tissues [22]. Our study showed that BMI1 was highly expressed in IDC tissue compared to normal tissue. A study by Althobiti et al. demonstrated that high BMI1 expression was associated with better prognosis in patients with luminal ER-positive breast cancer, but was associated with a poorer survival rate in patients with the triple-negative subtype. In patients with the HER2-positive subtype, there was no significant difference in survival [39]. Another study reported that high BMI1 expression could promote tamoxifen resistance in ER-positive breast cancers, with a high recurrence rate [23]. Thus, controversy still exists regarding the relationship between BMI1 and survival in breast cancer. Our results showed that high BMI1 expression was significantly associated with ER and PR negativity. The survival analysis revealed that high BMI1 expression was correlated with poor DFS and DSS, especially in patients with the luminal subtype. In this study, the machine learning model that incorporated BMI1 increased the predictive accuracy of the survival rate.

BMI1 has been actively investigated in various malignancies. A study by Proctor et al. showed that high BMI1 expression was observed in neoplastic pancreatic tissue, such as pancreatic intraepithelial neoplasias and pancreatic cancer cell lines [13]. In studies on liver disease, BMI1 was found to be highly expressed in hepatocellular carcinoma [14,15]. A follow-up study demonstrated that knockdown of BMI1 increased sensitivity to 5-FU treatment in hepatocellular carcinoma [16]. In a study of prostate cancer, high BMI1 expression was found to be correlated with unfavorable factors, such as a high Gleason score and extraprostatic extension with prostate-specific antigen recurrence [17]. A study by Yu et al. demonstrated that BMI1 expression was higher in endometrial adenocarcinoma tissues than in normal tissues and that high BMI1 expression was correlated with a poor survival rate [18]. A study of hematologic malignancies reported that BMI1 was highly expressed in patients with acute myelogenous leukemia or chronic myelogenous leukemia compared to the normal control group [19]. In a study of lung cancer, BMI1 overexpression was correlated with large tumor size, poor differentiation, distant metastasis, and worse survival in non-small-cell lung cancer (NSCLC) [40]. However, another study revealed that a high BMI1 mRNA level in blood was correlated with better survival in patients with advanced NSCLC [41]. In another study comparing BMI1 expression between small-cell lung cancer (SCLC) and NSCLC, BMI1 was found to be more highly expressed in SCLC than in NSCLC. In addition, BMI1 expression was not found in normal lung tissues [42]. Moreover, Li-Bing et al. reported that BMI1 was overexpressed in nasopharyngeal carcinoma cell lines compared with normal nasopharyngeal epithelial cells [20].

Antitumor immune activity could play an important role in the treatment of various solid tumors, including breast cancer [4]. The antitumor immune response can be enhanced by controlling CD8+ T cells [5]. We identified gene sets associated with of the decreases in CD8+ T cells and CD4+ T cells in the context of high BMI1 expression. A previous study demonstrated that PD-1/PD-L1 pathway inhibitors could potentially benefit patients with the metastatic triple-negative subtype [6]. We found that high BMI1 expression was related to decreased CD8+ T cell, CD4+ T cell, and TIL counts, as well as decreased expression of CD274 (encoding PD-L1). The decrease in both the CD8+ T cell count and CD274 expression suggests that the anticancer immune response is not regulated through PD-L1 signaling alone. A recent study of breast cancer reported that inhibition of BMI1 expression can eliminate BMI1-positive cancer stem cells, triggering endogenous immune responses, including CD8+ T cell expansion, in tumor cells [24]. Thus, we suggest that high BMI1 expression with a low CD8+ T cell count could promote breast cancer cell survival.

Telomeres are chromosomal ends containing tandem repeat DNA sequences—namely, TTAGGG in humans—and are essential structures for maintaining chromosomal integrity [43,44]. Somatic cells have a limited life span and stop dividing when a certain limited number has been attained; these cells then enter a phase of replicative senescence due to shortened telomeres [45,46]. Telomerase is a ribonucleoprotein complex that uses a region in its RNA component as a template for repeat synthesis, in which telomeric repeats are newly added to the 3’ end of the chromosome via RNA-dependent DNA polymerase [43]. In humans, hTERT (human telomerase reverse transcriptase), a subunit of human telomerase, is a 127-kDa protein whose function closely correlates with the control of telomerase activity [47]. In various types of human malignancies, telomerase activity has been investigated and detected [48,49,50,51]. In breast cancer, high levels of hTERT mRNA were found to be correlated with a poor clinical outcome [48]. A study by Goberdhan P et al. demonstrated that overexpression of BMI1 induced activation of hTERT transcription and telomerase activity, therefore enhancing the immortalization of mammary epithelial cells [52]. Since then, the same results have been reported for prostate and nasopharyngeal cancer [20,53]. In a recent study, the authors explored hTERT mRNA expression levels in breast cancer samples with a panel of 30 known stem cell marker molecules; in this panel, there was a significant association between hTERT and BMI1 expression in cancerous tissue [54].

This study has several limitations that should be acknowledged. First, neither the cross-sectional study nor in silico analyses of the TCGA database showed continuous relationships over time. Therefore, arriving at a definite conclusion is difficult. Second, molecular mechanisms were not shown using in vitro and/or in vivo experiments in our study. We should consider that in vitro and/or in vivo studies of CD8+ T cells, CD4+ T cells and PD-L1 are needed for further evaluation. Third, the pharmacokinetics observed in breast cancer cell lines may be highly different from those observed in patients, where various pharmacodynamic properties are affected by the disease status, microenvironment, and immune status. Fourth, there was no statistical relationship between BMI1 expression and vascular invasion in our study. Nevertheless, the patients with vascular invasion showed a higher proportion in the high BMI1 expression group (17.7%) compared to the low BMI1 expression group (14.7%). For these reasons, we could consider the heterogeneity of breast cancers with different molecular expressions. The function of the stem cell marker BMI1 may differ depending on different types of malignancy. In gastric cancer, BMI1 was reported to increase stem cell-like properties [55]. In endometrial cancer, high BMI1 expression was frequently observed in patients without vascular invasion [56].

In summary, this study demonstrated that high BMI1 expression was associated with a decreased CD8+ T cell count. The reduced antitumor immune response implies that high BMI1 expression could promote cancer cell survival in patients with breast cancer, especially those with IDC of the luminal subtype. Pathway-based network analysis revealed significant relationships between BMI1 and T cell receptor signaling and the inflammatory response pathway. This finding suggests that high BMI1 expression could be a therapeutic target in breast cancer. Our results will contribute to the design of future experimental studies and drug development for patients with breast cancer.

## Figures and Tables

**Figure 1 jpm-11-00739-f001:**
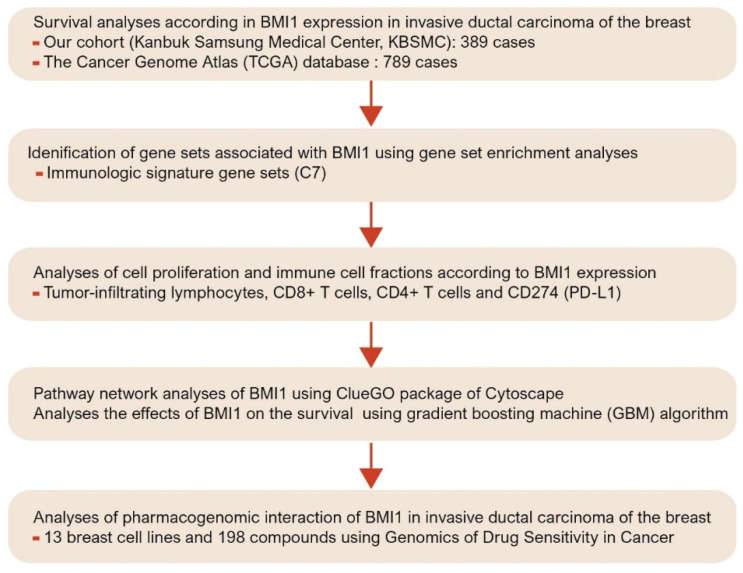
A schematic diagram depicting the analysis pipeline of this study.

**Figure 2 jpm-11-00739-f002:**
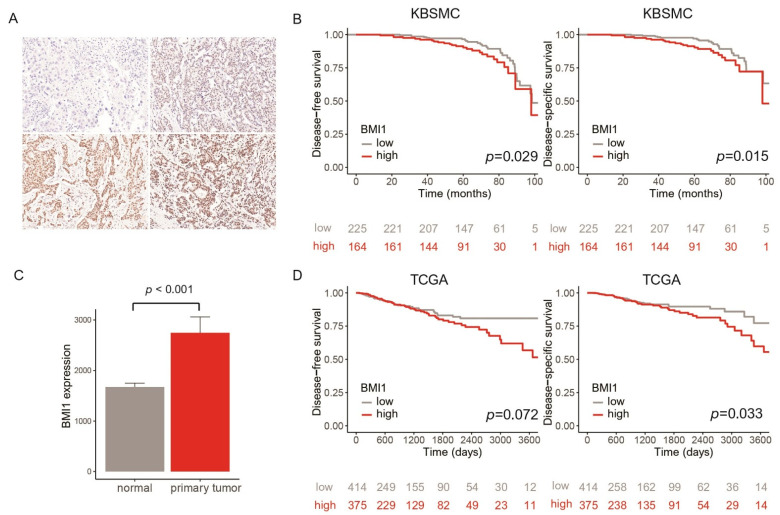
(**A**) The intensity of staining was scored as negative (left top), weak (right top), moderate (left bottom), or strong (right bottom) (original magnification 200×). (**B**) Survival analyses of the Kangbuk Samsung Medical Center (KBSMC) cohort: High BMI1 expression was associated with poor disease-free and disease-specific survival (*p* = 0.029 and 0.015, respectively). (**C**) Bar plots of The Cancer Genome Atlas (TCGA) cohort data: BMI1 expression was higher in primary tumors (*p* < 0.001). (**D**) Survival analyses of the TCGA cohort: high BMI1 expression was associated with worse disease-free and disease-specific survival (*p* = 0.072 and 0.033, respectively).

**Figure 3 jpm-11-00739-f003:**
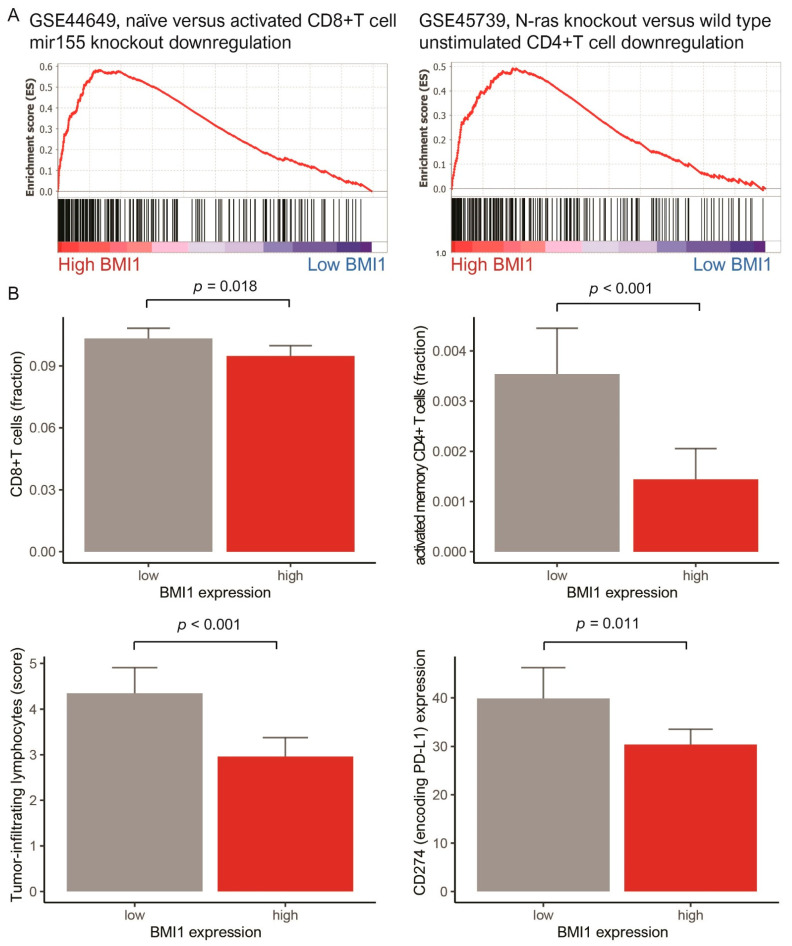
(**A**) Two immunologic gene sets associated with high BMI1 expression: GSE44649, naïve versus activated CD8+ T cell, mir155 knockout downregulation and GSE44649, naïve versus activated CD8+ T cell, unstimulated CD4+ T cell downregulation. (**B**) Bar plots showing the relationship of BMI1 expression with the following parameters: CD8+ T cells, activated memory CD4+ T cells, tumor-infiltrating lymphocytes, and CD274 (*p* = 0.018, <0.001, <0.001, and =0.011, respectively) (error bars: standard errors of the mean).

**Figure 4 jpm-11-00739-f004:**
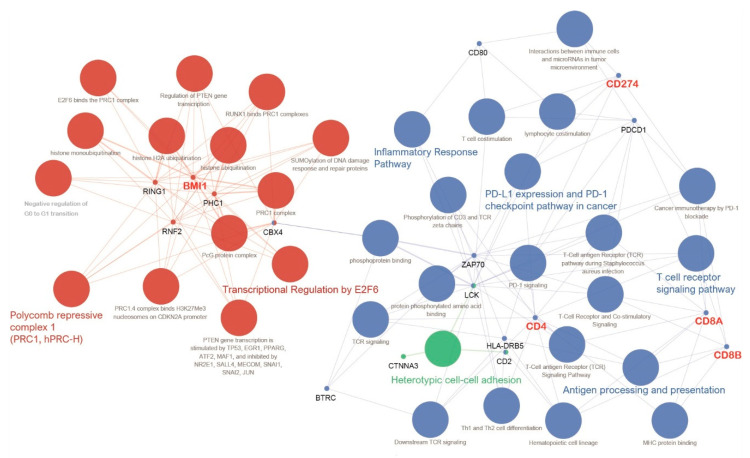
Grouping of networks based on functionally enriched Gene Ontology (GO) terms and pathways related to BMI1. The functionally grouped networks are linked to their biological functions, where only the most significantly enriched terms in the group are labeled: polycomb repressive complex 1 (PRC1, hPRC-H), transcriptional regulation by E2F6, inflammatory response pathway, PD-L1 expression and PD-1 checkpoint pathway in cancer, antigen processing and presentation, T cell receptor signaling pathway, and heterotypic cell-cell adhesion.

**Figure 5 jpm-11-00739-f005:**
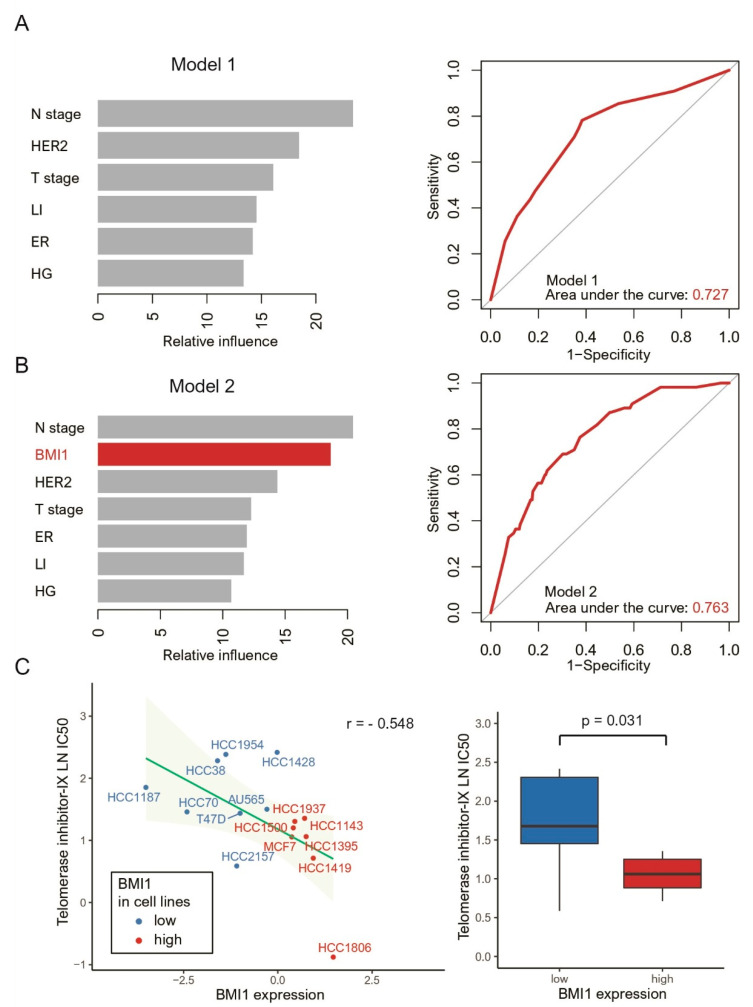
We applied supervised machine learning models for prognostic prediction using a gradient boosting machine (GBM). The covariates included the confounding factors ((**A**) Model 1; T stage, N stage, histological grade (HG), lymphatic invasion (LI), estrogen receptor status (ER), and Human Epidermal growth factor Receptor 2 status (HER2) versus (**B**) Model 2; BMI1, T stage, N stage, HG, LI, ER, and HER2) and their relative importance using survival. Receiver operating characteristic curves for the GBM models were generated based on a multivariate Gaussian model. (**C**) Pearson correlations (left) and box plots (right) showing the natural log half-maximal inhibitory concentration (LN IC50) of telomerase inhibitor IX, the most potent anticancer drug against breast cancer cell lines with high BMI1 expression.

**Table 1 jpm-11-00739-t001:** Clinicopathological parameters of patients in the KBSMC cohort stratified by BMI1 expression.

Parameter	BMI1 Expression		*p* Value
	Low (*n* = 225), *n* (%)	High (*n* = 164), *n* (%)	
Age			
≤55 y	161 (71.6)	121 (73.8)	0.627
>55 y	64 (28.4)	43 (26.2)	
T stage			
1	102 (45.3)	84 (51.2)	0.189 ^a^
2	105 (46.7)	62 (37.8)	
3	18 (8%)	18 (11)	
N stage			
0	154 (68.4%)	107 (65.2%)	0.758 ^a^
1	37 (16.4%)	34 (20.7%)	
2	25 (11.1%)	17 (10.4%)	
3	9 (4.0%)	6 (3.7%)	
Histological grade			
1	39 (17.3)	27 (16.5)	0.675 ^a^
2	113 (50.2)	90 (54.9)	
3	73 (32.4)	47 (28.7)	
Lymphatic invasion			
Negative	178 (79.1%)	131 (79.9%)	0.954
Positive	47 (20.9%)	33 (20.1%)	
Vascular invasion			
Negative	191 (85.3)	135 (82.3)	0.433
Positive	33 (14.7)	29 (17.7)	
ER status			
Negative	119 (52.9%)	48 (29.3%)	<0.001
Positive	106 (47.1%)	116 (70.7%)	
PR status			
Negative	152 (67.6)	91 (55.5)	0.015
Positive	73 (32.4)	73 (44.5)	
HER2 status			
Negative	147 (65.3%)	106 (64.6%)	0.972
Positive	78 (34.7%)	58 (35.4%)	

ER, estrogen receptor; PR, progesterone receptor; HER2, human epidermal growth factor receptor 2. ^a^ Linear-by-linear association.

**Table 2 jpm-11-00739-t002:** Disease-free and disease-specific survival analyses of patients stratified by BMI1 expression.

Survival	Univariate ^a^	Multivariate ^b^	HR	95% CI	
**Disease-free survival**					
BMI1 (low vs. high)	0.029	0.028	1.934	1.076	3.475
T stage (1 or 2 vs. 3)	0.015	0.358	1.418	0.673	2.987
N stage (0, 1 or 2 vs. 3)	<0.001	0.002	3.006	1.500	6.023
Histologic grade (1 or 2 vs. 3)	0.052	0.806	0.924	0.492	1.734
Lymphatic invasion (negative vs. positive)	<0.001	0.026	2.027	1.089	3.774
ER/PR (positive vs. negative)	0.310	0.033	0.509	0.273	0.948
HER2 (negative vs. positive)	0.634	0.632	0.868	0.487	1.547
**Disease-specific survival**					
BMI1 (low vs. high)	0.015	0.019	2.139	1.132	4.039
T stage (1 or 2 vs. 3)	<0.001	0.106	1.894	0.873	4.109
N stage (0, 1 or 2 vs. 3)	<0.001	0.006	2.819	1.341	5.922
Histologic grade (1 or 2 vs. 3)	0.049	0.639	1.174	0.601	2.291
Lymphatic invasion (negative vs. positive)	<0.001	0.008	2.485	1.264	4.886
ER/PR status (positive vs. negative)	0.938	0.645	0.851	0.429	1.688
HER2 status (negative vs. positive)	0.845	0.924	0.970	0.515	1.825

ER/PR, estrogen receptor/progesterone receptor; HER2, human epithelial growth factor receptor 2, T and N stages based on the definitions of the American Joint Committee on Cancer, 8th edition. ^a^ Breslow test. ^b^ Cox proportional hazards model.

## Data Availability

The data presented in this study can be available on request from the corresponding author. The data are not publicly available due to privacy.

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
