# Peer review of "High BMI1 Expression with Low CD8+ and CD4+ T Cell Activity Could Promote Breast Cancer Cell Survival: A Machine Learning Approach"

_jpm, 2021, doi:10.3390/jpm11080739_

Round 1

Reviewer 1 Report

The manuscript by Chung et al., describe the correlation between BMI1 expression and breast cancer survival and the signature pathway in this correlation. In general, correlation between BMI1 expression and cancer survival is well-defined in many cancers, but not breast cancer. Several concerns are as follows:

  1. In Fig. 1B, the graph seems no correlation between disease-free survival and BMI expression, but p value is less 0.05?
  2. In Table 1, since it is reported that BMI plays a role in stemness, why no correlation is observed in vascular invasion of patients?
  3. In Fig. 2, it seems that correlation was observed between BMI expression and CD4 T cell, not CD8.

Author Response

Reviewer 1

Comments and Suggestions for Authors

The manuscript by Chung et al., describe the correlation between BMI1 expression and breast cancer survival and the signature pathway in this correlation. In general, correlation between BMI1 expression and cancer survival is well-defined in many cancers, but not breast cancer. Several concerns are as follows:

In Fig. 1B, the graph seems no correlation between disease-free survival and BMI expression, but p value is less 0.05?

Answer:

We used the Breslow test instead of the log-rank test. The log-rank test gives equal weight to all time points and stages. The Breslow method gives more weight to deaths at early time points. Considering that most of the patients who die from breast cancer relapse early after surgery, we believed the Breslow method was appropriate.

In Table 1, since it is reported that BMI plays a role in stemness, why no correlation is observed in vascular invasion of patients?

Answer)

As you pointed out, we have added content to the discussion section explaining limitations as follows: 

“Fourth, there was no statistical relationship between BMI1 expression and vascular invasion in our study. Nevertheless, the patients with vascular invasion showed a higher proportion in the high BMI1 expression group (17.7%) compared to the low BMI1 expression group (14.7%). For these reasons, we could consider the heterogeneity of breast cancers with different molecular expressions. The function of the stem cell marker BMI1 may differ depending on different types of malignancy. In gastric cancer, BMI1 was reported to increase stem cell-like properties (52). In endometrial cancer, high BMI1 expression was frequently observed in patients without vascular invasion (53).”

In Fig. 2, it seems that correlation was observed between BMI expression and CD4 T cell, not CD8.

Answer:

As you pointed out, the title was revised as follows: 

“High BMI1 Expression with Low CD8+ and CD4+ T Cell Activity Could Promote Breast Cancer Cell Survival: A Machine Learning Approach”

Reviewer 2 Report

1) The validation methodology performed is not clear. A flowchart of analysis framework could be useful.

2) It appears that the authors did not use cross validation for the stepwise selection of features. The features in the stepwise selection were selected only

with respect to the performance assessed on training dataset of hold-out set and not in the test dataset, until after the selection procedure was completed.

3) The authors could perform a grafted cross validation of the feature selection algorithm in order to obtain a more robust subset.

4) Moreover, in order to obtain a measure of variability of the results obtained, it would be advisable to carry out several validation rounds. Other machine learning algorithms could be evaluate.

5) References should be implemented. I recommend that you enter the following articles:

  • Fanizzi A et al. Hough transform for microcalcification detection in digital mammograms. Appl. Digit. Image Process. XL 2017, 10396, 41, doi:10.1117/12.2273814
  • Fanizzi A et al. Ensemble DiscreteWavelet Transform and Gray-Level Co-Occurrence Matrix for Microcalcification Cluster Classification in Digital Mammography. Appl. Sci. 2019, 9, 5388
  • Losurdo L et al. Radiomics Analysis on Contrast-Enhanced Spectral Mammography Images for Breast Cancer Diagnosis:a pilot study. Entropy 2019, 21, 1110

Author Response

Reviewer 2

1) The validation methodology performed is not clear. A flowchart of analysis framework could be useful.

Answer:

As recommended, we added new figure 1.

2) It appears that the authors did not use cross validation for the stepwise selection of features. The features in the stepwise selection were selected only with respect to the performance assessed on training dataset of hold-out set and not in the test dataset, until after the selection procedure was completed.

Answer:

Gradient boosting machine (GBM) is a machine learning method based on decision trees. R package of GBM automatically divides training set (70%) and test (30%) set randomly. The following R code was used (Note: Code for model optimization was not described below). 

library(gbm)

DF1<-read_excel("our data for machine learning.xlsx", col_names = TRUE, sheet = 1)

set.seed(123)

DF1_fit_gbm <- gbm ( formula = death ~ T stage +N stage + Histological grade + Lymphatic invasion + ER + BMI1,   data = DF1,   var.monotone = NULL,   n.trees = 16,   interaction.depth =5,   n.minobsinnode = 10,   shrinkage = 0.2,    bag.fraction = 0.8,   train.fraction = 0.7,   cv.folds = 1000,   keep.data = TRUE,   verbose = TRUE,   n.cores = NULL)

3) The authors could perform a grafted cross validation of the feature selection algorithm in order to obtain a more robust subset. 

Answer:

As you pointed out, we plan to perform future studies by applying a better algorithm.

4) Moreover, in order to obtain a measure of variability of the results obtained, it would be advisable to carry out several validation rounds. Other machine learning algorithms could be evaluate.

Answer:

It is thought that a larger number of patients will be required for validation using machine learning. We have only used machine learning based on GBM, but we plan to use other machine learning platforms for future research.

5) References should be implemented. I recommend that you enter the following articles:

Answer: 

As recommended, we added the references.

Fanizzi A et al. Hough transform for microcalcification detection in digital mammograms. Appl. Digit. Image Process. XL 2017, 10396, 41, doi:10.1117/12.2273814

Fanizzi A et al. Ensemble DiscreteWavelet Transform and Gray-Level Co-Occurrence Matrix for Microcalcification Cluster Classification in Digital Mammography. Appl. Sci. 2019, 9, 5388

Losurdo L et al. Radiomics Analysis on Contrast-Enhanced Spectral Mammography Images for Breast Cancer Diagnosis:a pilot study. Entropy 2019, 21, 1110

Reviewer 3 Report

The authors demonstrated that high BMI1 levels are related to poor prognosis in breast cancer, using a machine-learning approach
The paper sounds very interesting and the identification of the telomerase inhibitor IX, a potential targeted therapy against breast cancer cell lines with high BMI1 expression, gives a further useful practical perspective to this study. 

I have only some minor concerns:

  1. The introduction is too general. For example when the authors use the term "Breast Cancer", they refer only to breast carcinomas. The mammary gland may be affected by several neoplasms, for example of the mesenchymal-type (Magro G. et al.; doi: 10.32074/1591-951X-31-19.). A brief mention to the fact that the breast may harbour different type of neoplasms should be added to the introduction.
  2. When authors mention different prognostic factors of breast cancer, they should expand the number of factors reported. For example a mention of additional prognostic factors, such as macroH2A (Broggi G. et al.; doi: 10.3389/fonc.2020.01519) and ITGA3 (Li et al; doi: 10.3389/fonc.2021.658547) should be included to this manuscript.

Author Response

Comments and Suggestions for Authors

The authors demonstrated that high BMI1 levels are related to poor prognosis in breast cancer, using a machine-learning approach

The paper sounds very interesting and the identification of the telomerase inhibitor IX, a potential targeted therapy against breast cancer cell lines with high BMI1 expression, gives a further useful practical perspective to this study.

I have only some minor concerns:

The introduction is too general. For example when the authors use the term "Breast Cancer", they refer only to breast carcinomas. The mammary gland may be affected by several neoplasms, for example of the mesenchymal-type (Magro G. et al.; doi: 10.32074/1591-951X-31-19.). A brief mention to the fact that the breast may harbour different type of neoplasms should be added to the introduction.

Answer:

As recommended, we added more content to the introduction section as follows: 

“In breast cancer, there are different histological types of malignancy such as ductal carcinoma, lobular carcinoma, medullary carcinoma and micropapillary carcinoma. As mesenchymal-type, breast cancer originating from stromal cells may occur rarely (2).”

When authors mention different prognostic factors of breast cancer, they should expand the number of factors reported. For example a mention of additional prognostic factors, such as macroH2A (Broggi G. et al.; doi: 10.3389/fonc.2020.01519) and ITGA3 (Li et al; doi: 10.3389/fonc.2021.658547) should be included to this manuscript.

Answer:

As recommended, we added the content in the introduction section as follows: 

“There are several biomarkers that have been discussed as prognostic factors such as JAK1, ITGA3 and MacroH2A1 in relation to the immune activity of breast cancer (9-11).”

Reviewer 4 Report

This well-written paper by Yumin Chung, entitled, “High BMI1 Expression with Low CD8+ and CD4+ T Cell Activity Could Promote Breast Cancer Cell Survival: A Machine Learning Approach”.  This close look at the expression of the BMI1 gene (B cell -specific Moloney murine leukemia virus integration site 1), in order to investigate the survival prediction model through a gradient boosting machine approach incorporating it, is well described.  The patient pools used in this study are quite impressive and the parameters evaluated are excellent. Both the tumor tissue microarray construction and evaluation of immunohistochemical staining are very well described by the authors. The summary captures the findings of this study in that it demonstrates that high BMI-1 oncogene expression was very much associated with a CD8+ T cell count decrease.  These decrease could be taken to mean this reduction could help in promoting cancer cell survival in patients that possess high BMI-1 expression. 

Round 2

Reviewer 1 Report

In the question 3, the authors directly change the title by adding CD4. It seems that the authors did not check whether it is CD8 or CD4 plays a role in this model.

Reviewer 2 Report

thank you very much for accepting my suggestions. In my opinion the work is now ready to be published